# MncR: Late Integration Machine Learning Model for Classification of ncRNA Classes Using Sequence and Structural Encoding

**DOI:** 10.3390/ijms24108884

**Published:** 2023-05-17

**Authors:** Heiko Dunkel, Henning Wehrmann, Lars R. Jensen, Andreas W. Kuss, Stefan Simm

**Affiliations:** 1Institute of Bioinformatics, University Medicine Greifswald, Walther-Rathenau Str. 48, 17489 Greifswald, Germany; 2Department of Biosciences, Molecular Cell Biology of Plants, Goethe University, 60438 Frankfurt am Main, Germany; 3Human Molecular Genetics Group, Department of Functional Genomics, Interfaculty Institute of Genetics and Functional Genomics, University Medicine Greifswald, 17475 Greifswald, Germany

**Keywords:** ncRNA, machine learning, transcriptomics, convolutional neural networks, deep learning

## Abstract

Non-coding RNA (ncRNA) classes take over important housekeeping and regulatory functions and are quite heterogeneous in terms of length, sequence conservation and secondary structure. High-throughput sequencing reveals that the expressed novel ncRNAs and their classification are important to understand cell regulation and identify potential diagnostic and therapeutic biomarkers. To improve the classification of ncRNAs, we investigated different approaches of utilizing primary sequences and secondary structures as well as the late integration of both using machine learning models, including different neural network architectures. As input, we used the newest version of RNAcentral, focusing on six ncRNA classes, including lncRNA, rRNA, tRNA, miRNA, snRNA and snoRNA. The late integration of graph-encoded structural features and primary sequences in our *MncR* classifier achieved an overall accuracy of >97%, which could not be increased by more fine-grained subclassification. In comparison to the actual best-performing tool ncRDense, we had a minimal increase of 0.5% in all four overlapping ncRNA classes on a similar test set of sequences. In summary, *MncR* is not only more accurate than current ncRNA prediction tools but also allows the prediction of long ncRNA classes (lncRNAs, certain rRNAs) up to 12.000 nts and is trained on a more diverse ncRNA dataset retrieved from RNAcentral.

## 1. Introduction

Besides the protein-coding messenger RNAs (mRNAs), non-coding RNA (ncRNA) classes have become more important in life science research, based on their various functionalities [1]. Even if the abundance of many ncRNA classes is quite low in comparison to rRNAs (~80%) and tRNAs (~10–15%), they are very important for essential regulatory processes, including cell homeostasis [2]. Based on their diversity in size, structural features, as well as sequential features, ncRNAs perform a variety of different tasks within the cell and can be further classified into RNA families such as miRNAs, rRNAs and snoRNAs [3]. In general, ncRNAs can be grouped mainly into housekeeping and regulatory ncRNAs (Figure 1) [4].

Among the housekeeping ncRNA families, ribosomal RNAs (rRNAs) and transfer RNAs (tRNAs) have been especially intensively analyzed as they are both involved in translation by either being structural elements in the backbone of the ribosome [5] or in transporting amino acids to the ribosome [6]. Short (<300 nts) nuclear RNAs (snRNAs) and short nucleolar RNAs (snoRNAs) are both involved in processing and modifying RNA sequences and can be split into several subgroups based on more specific functions. SnoRNAs can be divided into three main groups, H/ACA-Box, C/D-Box and scaRNA. The first group, H/ACA-Box snoRNAs, converts uridine residues to pseudouridine in rRNA and snRNA [7]. C/D-Box snoRNAs perform 2′O methylation of rRNAs [8]. Lastly, scaRNAs (small Cajal-body-associated RNA) can contain either of the two boxes or even have both boxes, meaning they can perform similar tasks to the other types of snoRNA [9]. SnRNAs—in contrast to this—are part of the spliceosome and can be subdivided by their specific position within it (U1, U2, U4, U5, U6, U11, U12). Within the spliceosome, they are responsible for catalyzing the splicing of pre-mRNA [10].

Short ncRNAs, such as microRNAs (miRNAs), small interfering RNAs (siRNAs) and PIWI-interacting RNAs (piRNAs), on the other hand, are assigned to the category of regulatory ncRNAs and play a role in transcriptional and translational regulation [11]. One very broad regulatory ncRNA family is so far quite heterogeneous and summarized under the term long non-coding RNAs (lncRNAs) [4]. In general, all ncRNAs, except rRNAs above a length of 200 nts, are at a stage assigned to this group. Additionally, subclasses such as long intergenic (linc) RNAs and antisense lncRNA are defined based on their location in the genome [11]. Apart from these ncRNA families, other classes of RNAs divided into cis- and trans-acting elements, regions such as IRES (Internal Ribosome Entry Site) or riboswitches, can perform regulatory functions within the cell, but are normally defined as regions of non-coding DNA, which regulate the transcription of neighboring genes [12,13]. Finally, different classifications and more fine-grained sub-classifications of RNA elements (cis and trans) and ncRNAs have been determined over the last years [14] due to their increased importance for medical and biological research [4].

In addition, during recent years, ncRNA research has become more and more important in the field of medical science and health care, as ncRNAs were linked with several diseases, including cancer, dementia and diabetes [15,16,17]. Especially for cancer diagnostics and therapeutic approaches, ncRNA families such as miRNAs or lncRNAs are being investigated [18]. For instance, H19, an lncRNA that also acts as a precursor miRNA, was found to have elevated expression levels in many types of cancer [19]. Certain miRNAs have also been found to be present in greater amounts in prostate cancer patients, including miR-21, whose plasma levels have been found to be significantly different between healthy subjects and prostate cancer patients [20]. As some miRNAs play an important role in tumor growth and cancer suppression [21], one research field is trying to use ncRNAs and especially miRNAs for treating cancer through the introduction of synthetic miRNAs [22]. Using so-called “miRNA replacement therapy”, miRNA let-7, which acts tumor-suppressively and has been found to be downregulated in cancers, has been the target of treatment using let-7 miRNA mimetics [23].

As research into ncRNAs in medical applications is currently mainly focused on specific ncRNA classes as well as individual ncRNAs, it is important to develop robust approaches for ncRNA classification to identify putative new candidates and allow a clear assignment. Since 2003, more and more publications have focused on the identification of new classes of ncRNAs based on either sequential, structural or functional information, for which reason “computational RNomics” for genome-wide annotation of RNAs became important [24]. Classification of short and long ncRNA classes can help to further characterize potential new ncRNAs found in high-throughput sequencing [25] and already assign them in a putative functional context without time-consuming in vitro or in vivo experiments. Besides the long and small ncRNA categories, the assignment of new ncRNAs to classes such as tiRNAs, siRNAs or miRNAs allows pre-filtering for the regulatory functionality in relation to diseases such as cancer [15] as well as their putative functionality for individualized therapies [26]. The development of next-generation sequencing (NGS) [27] applications has led to a very fast increase in the possibility of generating large amounts of data pertaining to expressed RNAs and partial RNA sequences. Therefore, machine learning (ML) approaches have been developed to improve the classification of RNA sequences based on primary sequence and secondary structure. There are a variety of approaches available for the differentiation between coding and non-coding RNA, including CPC2 [28] as well as tools for the identification of individual ncRNA classes, including LncDC [29] for differentiating lncRNAs from mRNAs. More recently, the rapid amount of increasing ncRNA sequences due to RNA-Seq experiments allowed the development of tools focusing their training on just a specific set of species such as microalgae (mSRFR [30]) or a taxonomic rank such as Viridiplantae (NCodR [31]) to reach higher accuracy. Specifically, the differentiation between multiple ncRNA classes within one sample may be of interest among a set of sequences, as this has the potential to improve the detection of ncRNAs in non-model organism genomes as well as the annotation of contigs of RNA-Seq samples in the future. For this reason, the classification of multi-class ncRNAs is of major importance and has evolved from multiple sequence alignment over simple machine learning classifiers (GraPPLE [32], RNAcon [33]) and deep learning artificial intelligence (AI) approaches, including nRC [34], ncRFP [35], ncRDeep [36] and ncRDense [37] up to natural language processing methods (NLP) such as ncRNLP [38]. The first tools to aid in the prediction of ncRNA types were GraPPLE (2009) and RNAcon (2014), which focus solely on the predicted secondary structure. Both approaches use a secondary structure as a mathematical graph, based on which they calculate 20 graph properties such as average path length or variance of closeness centrality. The difference between both tools lies in the ML classifier, as GraPPLE uses a support vector machine [39], while RNAcon uses a random forest [40] algorithm. Since then, the ncRNA classification has been mainly based on neural networks (NN) [41] as ML classifiers, which try to use the whole primary sequence and/or secondary structure for their prediction input. Common for all these approaches so far is the limitation of the input to a maximum of 750 input values (sequence length). In 2020, several deep learning approaches were released and tested on the Rfam database release of 2017 such as, for example, ncRNA_deep [42], which focuses on the prediction of specific Rfam families using a convolutional neural network (CNN), or ncRFP [35] and ncRDeep [36], which use a long short-term memory (LSTM) neural network or CNN to predict specific ncRNA and cis-regulatory element classes. All of them use one-hot encoded primary sequences of up to 200 nucleotides (nts), 500 nts or 750 nts in length as input, respectively. In 2021, the updated version of ncRDeep, called ncRDense [37], added a second CNN for predicting ncRNA classes using the secondary structure in dot-bracket notation [43] and merged both predictions in a last convolutional layer, thus combining both structure and sequence information, comparable to the 2017 released nRC [34]. The difference between nRC and ncRDense is the time point at which structure and sequence information is combined. nRC combines both information layers directly before usage as input, creating a binary encoded graph encoding vector representation using the MoSS algorithm [44] to connect specific secondary structure elements to the sequence. ncRDense inputs dot-bracket notation and sequence into two separate branches of the net and concatenates both afterward. In the latest benchmark of these tools, ncRDense had the highest overall F1-score with ~0.95 [37], using the Rfam (version 9.0 to 13.0 [45]). The NLP approach ncRNLP, published in 2023, focuses on the detection of small ncRNA classes using k-mer sequences as words [38].

At stage ncRNA, predictors have a maximum input length of 750 nts, which is not allowing to predict ncRNA classes such as rRNAs and miRNAs (pri-miRNAs > 10,000 nts [46]), including long and small ncRNA sequences leading to false positive classifications, even when using separated machine learning models specialized on long or small ncRNAs. Furthermore, the class of lncRNAs [47] can exceed the length of 750 nts, but in the current definition, the minimum length is 200 nts, which is also problematic in covering this class into a specific long or small ncRNA classifier. In addition, ncRNA classes such as snRNAs [10] are not included as separate individual classes so far or not balanced based on their heterogeneity in sequence length, such as miRNAs (mature miRNAs and pre-miRNAs) and rRNAs (small and large subunit). For this reason, we created a new training and validation dataset as well as an updated benchmark dataset based on RNAcentral [48] besides the existing benchmark dataset on the Rfam [49]. To investigate the prediction capacity of ncRNAs, we implemented and optimized several ML models and their inputs to tackle the high heterogeneity of ncRNA classes in sequence length and importance of sequence motifs and secondary structures. In this study, our models focus on the discrimination between six major ncRNA classes (snoRNA, snRNA, rRNA, tRNA, miRNA and lncRNA) occurring throughout eukaryotes. We compared the prediction accuracy of these ncRNA classes first on whole sequence-encoded information, weighted graph-encoded information or structurally encoded information. By late integration, our ML-model *MncR* (Merged_ncRNAclassifier) combining a CNN for sequence encoding information and a fully connected feed-forward artificial neural network (from now on in this article abbreviated with ANN) based on weighted graph encoding was benchmarked based on overall and single-class predictions. *MncR* was benchmarked against the current best ncRNA classification tool ncRDense [37] using two test datasets—one from RNAcentral (2022) and one from Rfam (2017).

## 2. Results

### 2.1. Increased Accuracy for ncRNA Prediction Combining Graph Vector Features and Primary Sequences

As the detection of ncRNAs and their function becomes more and more important [1], their classification and correct prediction are major tasks. During the last decades, different ncRNA prediction tools have been implemented using AI, relying either on primary sequence alone (ncRFP, ncRDeep) or on the combination of secondary structure and primary sequence (nRC, ncRDense). As ncRNA classes vary strongly in their length, and current tools are focusing more on short ncRNAs, we wanted to investigate the prediction performance for short ncRNAs, such as miRNAs, snRNAs, tRNAs and snoRNAs, while adding long ncRNA classes such as rRNAs and lncRNAs. Furthermore, the selection of these ncRNA classes is based on their structural or sequential motifs required for fulfilling their functions [4]. To get insights into the overall prediction capacity of sequence and structure information, as well as their combinations, we implemented and optimized four ML models using fully-connected feed-forward ANNs and CNNs, as well as late integration of both to predict six ncRNA classes: miRNA, snRNA, snoRNA, tRNA, rRNA and lncRNA. As input for the ML models, we created balanced ncRNA training, validation and test datasets from the RNAcentral database. Of the four ML models, ten-fold cross-validation consistently showed the best performance for the merged late integrated model combining graph encoding and sequence encoding (*MncR*) compared to the CNN models for solely structural encoding (*StrEnc*), primary sequence encoding (*SeqEnc*) and the ANN of graph structure encoding (*GrEnc*) (Figure 2).

After epoch five, *GrEnc* consistently showed the highest validation loss and—comparing the overall lowest median validation loss for the other three models—a loss below 0.2, with the lowest for the *MncR* model (Figure 2a). In summary, all four models achieved a high accuracy (F1-score > 0.94), and for each model, recall and precision values were within 0.001 of each other, with precision always being marginally higher than recall. In the direct comparison using precision and recall of the best ML model for each encoding as well as the accuracy scores F1 and MCC, we observed a higher F1-Score of the *SeqEnc* model by ~0.02 as compared to the *GrEnc* model, while *StrEnc* improved the F1-score of the *SeqEnc* model even further by ~0.003 (Figure 2b). Interestingly, the combination of the two information layers *SeqEnc* and *GrEnc* in the *MncR* model led to an overall improvement of prediction accuracy by ~0.005, leading to an F1-score of 0.979 and an MCC of 0.974. From our balanced test dataset, 5517 of the 6001 sequences were correctly predicted by all four models. Furthermore, 5970 of the 6001 sequences were correctly predicted by at least one of the four models, of which 48 were correctly identified by only one model (Figure 2c). Focusing on the two information encoding methods involving the structure of the sequence, we observed that 12 structural and 8 graph sequences were correctly classified by only one of them. Notably, the *MncR* model was able to correctly predict many of the ncRNAs previously wrongly predicted by the *GrEnc* or *SeqEnc* model and also covered most of the sequences correctly predicted by the *StrEnc* model, even though this information layer was not explicitly added in the *MncR* model. The *MncR* model falsely classified overall only 2.15% of the full dataset.

As these general accuracies do not directly give information about the individual ncRNA family prediction accuracy, we analyzed the prediction capacity of the four implemented NN models for the individual ncRNA families (Figure 3).

In the scores, we observed that the *MncR* model had the highest F1-scores for lncRNAs, snRNAs and snoRNAs but, in all other cases, performed slightly worse than the *SeqEnc* and *StrEnc* models (Figure 3a). Most notably, the *GrEnc* model was relatively bad at predicting the ncRNA classes rRNA, snRNA and snoRNA compared to the other models (0.02–0.04 lower F1-score than the second lowest). The *SeqEnc* model was less accurate in predicting miRNAs (0.94) compared to the *StrEnc* model (0.97), which, vice versa, had problems with the correct prediction of lncRNAs (*SeqEnc*: 0.97; *StrEnc*: 0.96). This trend where the *SeqEnc* was better for lncRNA and worse for miRNA than the *StrEnc* was also seen in the comparison of recall and precision, meaning that there was not a big difference in false positive or false negative prediction rates within the models (Figure 3b,c). With respect to ncRNA family classification, snoRNA has the lowest precision in all models (<0.951). In contrast, the recall scores of the best prediction were within a range between 0.955 and 0.977. The highest F1-scores were achieved for the classes tRNA (all models > 0.985, all except *GrEnc* > 0.993) and rRNA (all models > 0.967, all except *GrEnc* > 0.986). This is also reflected in the recall for tRNA (all models >0.98, all except *GrEnc* over 0.99) and rRNA (all models > 0.97, all except *GrEnc* > 0.979). With respect to precision, the scores for tRNA were even higher (all models > 0.99, all models except *GrEnc* > 0.997), with the *StrEnc* model even achieving perfect precision for this class. For rRNA, the precision of the *GrEnc* model was 0.014 lower than those of the other models (*GrEnc*: 0.965, all others > 0.989), with the *StrEnc* model also achieving the highest precision at 0.994, 0.004 higher than that of the *SeqEnc* and the *MncR* model. Overall, while there were minor differences between the classes, the *MncR* model achieved good F1-scores for all ncRNA classes (>0.963) and had the highest overall scores of all models (Figure 3b).

Next, we wanted to investigate the wrongly assigned family-wise ncRNA sequences between our models to detect possible noise added by the different information layers for prediction (Appendix A). We analyzed the 484 sequences that had been wrongly predicted by any model and determined for which of these the *MncR* model was wrong, for how many samples all models predicted wrongly and for which only the *MncR* model failed to predict correctly (Table 1). Only 17 ncRNA sequences were wrongly assigned by the *MncR* model alone, including six labeled lncRNAs and between one and three from all other ncRNA families. In contrast, 29 sequences were falsely predicted by all four models. The *MncR* model had the highest fraction of wrong classifications for lncRNAs compared to the other ncRNA classes (40/87). The high recall for snoRNA in comparison to the other models (Figure 3a) is reflected in the lower number of false classifications for snoRNA (23/136). Given this information, we individually analyzed what sequences are among the false predictions (Appendix A). Most notably, of the 40 lncRNA sequences that were wrongly classified by the *MncR* model, 18 were correctly classified by *SeqEnc* and *StrEnc*. Ten of these sequences had the same prediction from the *GrEnc* and *MncR* models. For snoRNA, 60 out of the 136 sequences wrongly classified by any model were only misclassified by the *GrEnc* model. Twenty-six of these were classified as snRNA. For snRNA, only the *GrEnc* model falsely predicted 33 sequences, 24 of which were predicted as snoRNA.

As we could observe, some problems—at least in the *GrEnc* model—concerning the discrimination between snRNAs and snoRNAs, as well as between lncRNAs, rRNAs and miRNAs, we split the snoRNAs into the subclasses scaRNA, C/D-box and H/ACA-box. Furthermore, we wanted to analyze the influence of splitting ncRNA families such as miRNAs and rRNAs based on their length on the prediction capacity (Figure 4).

The fine-grained *MncR* model improved precision for C/D-Box snoRNAs to 0.98, while scaRNA remained at the same precision as the general class *MncR* model (0.95) and H/ACA-Box showed a slight decrease in precision (0.94). Overall, we observed a big portion of scaRNAs (5%) being falsely assigned to the H/ACA-Box class. The recall for each snoRNA class in the fine-grained model (C/D-Box: 0.96; H/ACA-Box: 0.95; scaRNA: 0.94) was lower than in the general class model (snoRNA: 0.98). For miRNA, splitting the class into mature miRNA (~22 nts) and pre(cursor)-miRNA (>35 nts) led to perfect classifications for the mature miRNAs but a decrease in recall and precision for the pre-miRNAs (general classes miRNA: 0.97; fine-grained pre-miRNA: 0.95). The general class model already classified all mature miRNAs correctly (Appendix A) and was not influenced by the mixture of pre-miRNA and mature miRNA.

Splitting rRNA into short (5S, 5 8S) and long (28S, 25S, SSU, 23S) rRNAs leads to a slight decrease in recall (general classes: 0.99; fine-grained short: 0.97; long: 0.97). Precision evens out at 0.98 for long rRNA and 1 for short rRNA, compared to 0.99 in the general classes model. Furthermore, the unsplit ncRNA classes showed no indirect improvement in their prediction capacity. LncRNA showed an increase in recall (general classes: 0.96; fine-grained: 0.98) but a decrease in precision (general classes: 0.97; fine-grained: 0.96). For snRNA and tRNA, recall remained the same, but precision decreased by 0.01 to 0.98 for snRNA and 0.99 for tRNA, respectively. As splitting the classes led to no clear improvements in the overall prediction accuracy, the *MncR* model for the benchmark was based on the general six ncRNA classes.

### 2.2. MncR Model Outperforms Benchmarked ncRNA Classification Tool Based on Two Different Test Sets

After optimizing the different ML models and including several sequence information layers, we wanted to compare our *MncR* model to the current best ncRNA classifiers, such as ncRDense [37]. In their comparison to the tools ncRDeep [36], nRC [34], ncRFP [35] and ncRNA_deep [42], they used the benchmark Rfam test set from nRC [34]. In general, ncRDense outperformed all other mentioned ncRNA classifiers in accuracy, sensitivity, specificity and F1-score by at least 0.0019 to 0.0047 for the second-best tool, ncRNA_deep [42]. As current classifiers such as nRC, ncRFP, ncRDeep and ncRDense classify a mix of ncRNA major classes (tRNA, Ribozyme) and subclasses (precursor miRNA, 5S rRNA, 5.8S rRNA, C/D-Box, H/ACA-Box, scaRNA) as well as cis-regulatory elements (Riboswitch, Leader, IRES, Intron-gpI, Intron-gpII), we had to adjust the benchmark dataset for a fair comparison. A similar approach was used with ncRDense to use ncRNA_deep for comparison, which is based on Rfam family prediction. For this reason, we had to restrict the Rfam dataset [34] to selected sequences with labels existing in both ncRNA prediction models, including tRNA, snoRNA (ncRDense: scaRNA, C/D-box, H/ACA-box), miRNA and rRNA (ncRDense: 5S rRNA, 5.8S rRNA). Furthermore, we excluded sequences longer than 750 nts as it is not possible for ncRDense to generate a prediction for these. Lastly, we excluded all sequences with similarities above 90% to the training datasets (see Material and Methods). For the remaining four classes, we had 1125 sequences split into 401 snoRNAs, 184 miRNAs, 347 rRNAs and 193 tRNAs for our benchmark Rfam test dataset. In addition, our updated RNAcentral test set was also restricted by the same criteria, reducing the number of eligible sequences in the test set from 6001 to 2737 (523 miRNAs, 952 snoRNAs, 993 tRNAs and 269 rRNAs) (Figure 5).

In general, the *MncR* model outperformed ncRDense on the 1125 Rfam test set sequences in precision, recall, F1-score and MCC (Figure 5a, left). Overall, the prediction accuracy could be slightly improved in the range between 0.033 and 0.062 as compared to ncRDense. For the newly created RNAcentral benchmark test set, the differences were bigger, ranging from an improvement of 0.062 for the precision to 0.118 for the MCC (Figure 5a, right). To find out how well each model predicts certain ncRNA classes, we compared class-wise F1-scores on the Rfam test set and RNAcentral test set (Figure 5b). ncRDense had slightly higher F1-scores for tRNA (+0.008) and rRNA (+0.005), while our *MncR* model showed better performance for miRNAs (+0.154) and snoRNAs (+0.05). Especially miRNAs showed a drastic increase from 0.79 to 0.94, while for all other classes, both models were in the range between 0.92 and 0.99. We observed that of the 1125 sequences of the Rfam test set, 1019 could be correctly identified with both NN models (Figure 5c, left). More specifically, the *MncR* model was able to correctly identify 50 miRNA and 17 snoRNA sequences that ncRDense misclassified, as well as 4 rRNAs and 3 tRNA sequences. On the other hand, ncRDense classified seven miRNAs, nine rRNAs, three snoRNAs and four tRNAs correctly that were missed by our *MncR* model. Interestingly, nine sequences were incorrectly classified by both models, consisting of one miRNA, four rRNA and four snoRNAs (Appendix A). For the RNAcentral set, the *MncR* model had a higher F1-score for all four classes (Figure 5b, right). Especially for miRNA, snoRNA and rRNA, our *MncR* model achieved much higher scores with improvements of 0.225, 0.076 and 0.042, respectively, while the improvements for tRNA were smaller (0.009). Of the 2732 eligible sequences in the RNAcentral test set, both models correctly predicted 2414 (Figure 5c, right). Furthermore, 264 sequences were only correctly predicted by the *MncR* model, while 29 sequences were falsely predicted by the *MncR* model and correctly by ncRDense. In the test set, 30 sequences could not be identified by either of the two models.

## 3. Discussion

Our *MncR* model, using graph-encoded structural information late integrated with sequence information, is an ncRNA classifier that can predict six ncRNA classes (miRNA, snRNA, snoRNA, lncRNA, tRNA and rRNA) with a sequence length limitation of 12,000 nts (99.7% of sequences in RNAcentral). Previous multi-class ncRNA classifiers only allowed input lengths between the maximum lengths of 250 and 750 nucleotides. This multi-class ML model based on deep learning now also includes the classification of lncRNAs and ncRNA classes that contain long sequences such as rRNAs (including 16S, 18S, 23S, 25S and 28S) or miRNAs (including pre-miRNAs). In addition, we added snRNAs as a separate ncRNA class and the training set for miRNAs included a balanced amount of precursor and mature miRNAs, which so far has not existed in the Rfam training datasets used by previous models for ncRNA classification. To focus on major ncRNA classes, we excluded cis-regulatory elements such as IRES, Intron gpI + II, leader and Riboswitches [4] from the prediction. We also excluded the classes Ribozyme, piRNA (PIWI-interacting RNA) and siRNA (small-interfering RNA). Based on the dynamically and fast increasing amount of sequences from all kingdoms in publicly available databases (e.g., RNAcentral [48]), we created a new benchmark test dataset by increasing the number of ncRNA sequences and improving the balance between the six ncRNA families. While RNAcentral classifies some Ribozyme sequences as ncRNA, these belong primarily to insects and unicellular organisms and do not exist in all eukaryotes [48]. A second issue with Ribozymes is that nowadays, some are classified as lncRNAs, or vice versa, which would create an overlap between such classes [50]. For siRNAs in RNAcentral, the vast majority of ~98% belongs to only one organism, i.e., *A. thaliana*. Moreover, the helical double-stranded nature of their guide strand and passenger strand [51] cannot be compared with linear single-stranded RNAs. piRNAs were excluded as they mainly exist in mammalian cells [52].

The sequences from RNAcentral include sequences from 37 different expert databases, of which Rfam [40] (29.4%), ENSEMBL [53] (21.1%) and ENA [54] (20.6%) make up the largest portions in our dataset, while previous ncRNA classifiers only include sequences from the Rfam library [34]. In addition to the different class selection, our models are also based on much bigger datasets, including more samples per class (our model: ~10,000 samples per class; previous ncRNA classifiers: up to ~700 samples per class) as well as including all kingdoms (e.g., ~11% of our RNAcentral training sequences belong to Viridiplantae). Deep learning models of all kinds have been observed to show increased performance scaling with the number of training samples [55]. Previous classifiers have also included ncRNA subtypes such as H/ACA-Box, C/D-Box and scaRNA as individual RNA types instead of grouping them together as snoRNA. While there are differences between these three subtypes, they are biologically much more closely related in function and structure than, for instance, miRNAs and tRNAs [14]. Labeling them individually leads to unaccounted imbalances in a dataset, and training on imbalanced data usually causes the predictor to be stronger in predicting the majority class [56].

In earlier benchmarks for ncRNA classification, it could be shown that CNNs [36] outperform RNNs [35] using primary sequence information, having a higher true positive prediction rate [37]. For our implemented and optimized ML models, we focused for this reason on the different inputs based on sequence and structure information. The included weighted graph-encoding vectors derived from GraphProt in the *GrEnc* not only indicate the presence of certain features but also how often they occur. Additionally, the input vector of *GrEnc* includes ~5 times (32,768 features) more features than current ncRNA classifiers such as nRC (6483 features) [34], allowing the more fine-grained encoding of secondary structure. In general, the advantage of encoding the sequence structure as a graph to detect far-distanced motifs and connected structural elements [57] was not effective for the more accurate ncRNA classification in comparison to overall sequential information in the *StrEnc* model. Our *StrEnc* model is based on structure encoding by Pysster [58] using the sequential dot-bracket notation but did not perform better than the pure primary sequence in our *SeqEnc*. Interestingly, the combination of *GrEnc* and *SeqEnc* in our *MncR* seems to use the graph encoding to find far-distance motifs in combination with the sequential information to reach a higher accuracy (F1-score: *MncR*: 0.98) (Figure 2). In contrast to existing tools such as ncRDense, this late integration approach (established in heterogeneous multi-omics approaches [59]) of the *MncR* performed better on both benchmark datasets.

Focusing only on the overall prediction accuracy can lead to the conclusion that the sequence information alone performs better than the structural information (Figure 2). Besides the overall prediction accuracy, we observed different strengths and weaknesses of the single models regarding the single ncRNA classes (Figure 3). In general, we observed for five of the six ncRNA classes a better performance on the sequence alone; only the miRNAs were more accurately predicted using the structural information. Surprisingly, the *GrEnc* model did not outperform the *SeqEnc* model for snoRNAs, even though they have highly conserved secondary structure motifs but only very small conserved sequence motifs [60]. Additionally, while having a slightly lower recall (−0.006) than the *StrEnc* model, the *SeqEnc* model even had a higher precision (+0.007), which means that the introduction of the secondary structure using Pysster [58] did not change the prediction for snoRNAs. This may potentially be attributed to difficulties with the correct prediction of large internal loops by secondary structure prediction tools, which are found in all types of snoRNAs [60]. Large internal loops are not energetically stable, meaning minimum free energy predictors such as RNAfold [43] (used by Pysster [58]) and RNAshapes [61] (used by GraphProt) avoid them, which can add noise by artificial secondary structures to the data [62]. Aside from snoRNAs, we also find unexpected behavior for the class of miRNA, with the *SeqEnc* model performing worse than the *GrEnc* model regarding F1-score (−0.02). While none of the mature miRNAs are falsely classified (Appendix A), precursor miRNAs contain the mature miRNAs, meaning they contain highly conserved sequence motifs, which we would expect the *SeqEnc* model to be able to classify better than the *GrEnc* model [63]. At the same time, the secondary structure has also been found to be a good identifier for precursor miRNAs, which could explain the increased scores with the structure and graph features [64]. Based on our results, the hypothesis that ncRNA class prediction with a specific structure, such as snoRNAs, benefits from using just structural information could not be confirmed. Moreover, the single information (sequence or structure) shows tendencies to improve specific ncRNA classes, and the combination in the merged *MncR* combined both strengths. As the overall prediction accuracy from all of our four models was above an F1-score of 0.95, we only had problems classifying 31 sequences. For 11 of these, all four models agreed on the same false classification, including the tRNAs URS00021E9E8E_158383, URS00021942FF_158383 and URS0001BD2402_9606, where all four models predicted lncRNA. Nevertheless, based on sequence comparison on the Rfam, all three sequences showed only 6.5%, 6.1% and 4.1% similarity to tRNAs, respectively. This means that the predicted classification of lncRNA could be basically correct, as lncRNAs are known to contain tRNA-like subsequences at the 3′-end [65]. Similarly, one precursor miRNA was predicted as lncRNA by all models, which could be in line with previous findings, such as for lncRNA H19, which functions as a precursor to a miRNA (miR-675) [66]. By far, the largest groups of misclassifications were found in the classification of snoRNAs and snRNAs from the *GrEnc* model, where the model misclassified 60 snoRNAs (26 of which were assigned to the snRNA class, 17 lncRNAs, 9 rRNAs, 7 miRNAs and 1 tRNA) that the other three models identified correctly. Vice versa, the *GrEnc* model was the only one to misclassify 33 snRNAs, of which 24 were falsely labeled as snoRNA (four times labeled as lncRNA, three times as miRNA and two times as rRNA). These two ncRNA classes share many motif features, for example, the U3 C/D-Box snoRNA, which was originally classified as snRNA [67]. Interestingly, comparing the different ML models with each other gave new insights into the prediction capacity and thus may also lead to the reassignment of some of the sequences in RNAcentral. This must be further analyzed in the future. Specifically, the prediction of ncRNA classes with a clear conserved secondary structure, e.g., snoRNAs, tends to be worse by graph-encoding vector models as compared to pure primary sequence models (Figure 2c). As snoRNAs can be divided into three subclasses (C/D-Box, H/ACA-Box, scaRNA), all showing a specific secondary structure, current structure prediction tools such as RNAshapes [61] have difficulties predicting them accurately by minimum free energy [68]. The same phenomenon could be observed for the snRNAs, with our sequence model, which appears to be better at correctly identifying snRNAs, while the *GrEnc* model often incorrectly labels snRNAs as snoRNAs and vice versa. In contrast, more sequence-based ncRNAs, such as miRNAs and rRNAs, were more accurately and correctly predicted by the *GrEnc* model. For rRNAs, the secondary structure seems to allow better discrimination between lncRNAs and rRNAs as the ncRNA classes with the longest sequences. For miRNA, the secondary structure improves overall scores with the *GrEnc* model, outperforming the *SeqEnc* model regarding recall, precision and F1-score. This is surprising given the importance of the sequence for the function of miRNAs. All three models correctly identified all mature miRNAs, with the *SeqEnc* model being the only one to misclassify 39 sequences, of which 28 were mislabeled as snoRNA. For tRNA, only minor differences were found between the three models, with the *GrEnc* model being slightly lower and the *SeqEnc* model even classifying two more sequences correctly as compared to the *MncR* model. Lastly, lncRNA, as the fuzziest class of ncRNAs, contained 14 sequences, which could not be correctly predicted by any of the models. This could be due to the fact that the original definition of lncRNAs classified all non-coding RNAs longer than 200 nts that do not belong to any of the other classes as lncRNAs, regardless of function [14].

To exclude the possibility that our models wrongly assigned snoRNAs and snRNAs based on the distinct subtypes of snoRNAs, which are used in classifiers, such as nRC [34], ncRDense [37] or ncRDeep [36], we created and trained a more fine-grained *MncR* model with 10 classes (see Figure 4). In general, the split of snoRNAs in scaRNA, C/D-box and H/ACA-box decreased the recall for all subtypes, but this is partially attributed to misclassifications, e.g., scaRNA classified as H/ACA-Box and vice-versa, that previously had not been possible, because the general classes model does not differentiate between the two. ScaRNAs are snoRNAs having features characteristic of C/D-Box and H/ACA-Box snoRNAs (or both), and thus cross-talk between the classifications is to be expected [9]. Another problem is the low number of scaRNAs in comparison to C/D-box and H/ACA-box, leading to an imbalance within the snoRNA class, which can be crucial for deep learning methods at this stage [69]. Additionally, we split the ncRNA classes rRNA and miRNA into long and short to see if this approach has an influence on the prediction accuracy. This approach is, in principle, comparable to using two distinct classifiers for long and small ncRNAs with the advantage of giving the ML model a chance to also select from the additional ncRNA classes. For the rRNA, we observed no improvement in the split of long and small rRNAs but a slight decrease in the overall performance. For the miRNAs and pre-miRNAs split, we observed a decreased performance in the detection of pre-miRNAs and an increased performance on miRNAs. Overall, the sequence length-based categorization for miRNAs as pre-(cursor) and mature miRNAs, as well as for rRNAs as long and short rRNAs, showed no real improvement, if not even a slight decrease in prediction capacity. Furthermore, the inclusion of small and long ncRNA classes, even including the embedding, showed no decrease in the overall prediction accuracy with F1-scores above 0.95 (Figure 2 and Figure 4). Focusing on the falsely assigned sequences per ncRNA class, we observed that the few false positives in the *MncR* model were irrespective of the length of the ncRNA sequence.

In contrast to tools like LncDC [29], the focus of *MncR* is not to discriminate between protein-coding mRNAs and specific ncRNA classes, such as lncRNAs. Such tools are important to pre-filter datasets of RNA-seq sequences to exclude mRNAs—as those have partly similar features to lncRNAs—or to identify putative novel lncRNAs. Often the training to discriminate between mRNA and ncRNA has to be done individually for specific species, such as humans in LncDC [29] or taxonomic classes, e.g., microalgae in mSRFR [30]. The approach of mSRFR showed clearly that specific distinctions of sequence types between different kingdoms exist and have to be present in the training dataset to be learned by the ML model. The other universal sequence classifiers, such as *MncR* or ncRDense [37] want to achieve the detection of ncRNAs irrespective of special features of distinct species or taxonomic kingdoms. Tools such as mSRFR [30] or NCodR [31] focusing on a specific set of species increase the sensitivity for distinct features, e.g., additional variants of snRNAs in plants, which can be very short in length [70], but also decrease the possibility to be generalized and increase the chance of overfitting. In the case of NCodR, the publication showed a drastic increase in the F1-score by 30% compared to ncRDeep solely trained on Viridiplantae sequences [31]. To test MncR on a plant-specific dataset, we created a test set with 996 sequences (labeled with the current RNAcentral annotation) from the ~500m000 plant sequences of NCodR and classified them with *MncR,* achieving an F1-score of 0.85 (Appendix A). As *MncR* is trained on a dataset including ~11% of plant sequences, the model was still able to achieve high recall values for all types of plant ncRNAs (> 0.87) with the exception of snRNA (~0.40). Besides the internal comparison of different models using different input sources, we compared our *MncR* model to the currently best multi-class ncRNA classifier ncRDense [37], which is also based on deep learning. As ncRDense outperforms the other tools, ncRFP and nRC, by at least 0.17 (F1-score) and ncRDeep by 0.07, we wanted to compare our *MncR* model with ncRDense. Therefore, we used the classic Rfam test set created by the developers of nRC [34] and adjusted it based on the common ncRNA classes. *MncR* (F1-score: 0.9738) achieved higher accuracies than ncRDense with increased precision (+0.033), recall (+0.045) and F1-score (+0.04) (Figure 5). It was able to correctly classify 74 sequences, which ncRDense failed to classify, while ncRDense only correctly classified 23 sequences that our model could not correctly predict. The scores for ncRDense differed slightly from their publication, as we had to adapt the test set to not include any duplicates to either of the training sets of the tested models. In 2023, a new tool based on NLP called ncRNLP [38] trained and tested on the same Rfam dataset as ncRDeep and ncRDense were released but benchmarked only against ncRDeep. Both publications of ncRDense and ncRNLP compare themselves to ncRDeep (F1-score: ~0.88) and got similar benchmark results resulting in an improvement of the F1-score by 0.07 (ncRDense) to 0.09 (ncRNLP). Relating these benchmarks to our results allows us to conclude that our ML model *MncR* also performed slightly better than ncRNLP on the Rfam dataset. By checking the benchmark datasets, we observed that 347 sequences of the original Rfam benchmark dataset from 2017 are present in both the training and the test set of ncRDense, of which 131 belong to the four ncRNA classes we compared. When only testing the sequences of overlapping ncRNA classes, our *MncR* model overall outperforms ncRDense (Figure 5), mainly because of increased accuracy in miRNA prediction. Even more drastic is this result using our newly designed RNAcentral benchmark dataset. The biggest difference in the results can be found for miRNA, but also for the other ncRNA classes, tRNA, rRNA and snoRNA; we achieved better accuracies for our *MncR* model.

## 4. Materials and Methods

### 4.1. Creation of Training, Validation and Test Datasets

The dataset used for training as well as benchmarking our model is based on the RNAcentral database v18 [48] to create a balanced and wide-range dataset that includes sequences from six diverse ncRNA classes: rRNA, tRNA, miRNA, snRNA, snoRNA and lncRNA. All sequences had a minimum length of 15 nts and a maximum length of 11,922 nts. Partial sequences were excluded by including “NOT partial” in the search term. For balancing each ncRNA class, we downloaded equal amounts of subtypes for each ncRNA class (Table 2). For example, we included sequences from C/D-Box, H/ACA-Box and scaRNA for snoRNAs. In addition, we had to create a special criterion for miRNAs as this class is not fully divided into mature or precursor miRNAs, which can have a big influence on the classification. As not all miRNAs are divided into mature/precursor on RNAcentral, we selected 50% of the miRNAs to be below 31 nts, which we labeled as “mature”, and 50% to be above 39 nts (up to 6306 nts), which we labeled as “precursor”. For reducing redundant or nearly similar sequences within the individual datasets, we performed a clustering with CD-hit-est [71] among each of the ncRNA classes, removing sequences with a similarity larger than 90% using a word length of 7. Of the dataset, the biggest portion stems from the Rfam library [40] (29.4%), followed by Ensembl [53] and ENA [54] at 21.1% and 20.6%, respectively (Appendix A).

For the training and testing of our models based on the RNAcentral, we created one dataset including 10,000 sequences for each of the six ncRNA classes, equally distributed across the subtypes if possible (Table 2). Since not enough sequences were available for the snoRNA subtype scaRNA and the rRNA subtypes 25 s and 58 s, we included as many sequences as possible and then balanced our dataset across the other subtypes. For rRNAs (10,004) and snoRNAs (10,001), balancing across the subtypes led to negligibly small imbalances, and the remaining four classes lncRNAs, tRNAs, snRNAs and miRNAs, consisted of exactly 10,000 sequences for each class. In the beginning, the datasets for each ncRNA class were split into a train and validation (90/10 split for 9000 sequences per class) dataset as well as a test dataset (1000 per class). For the more fine-grained ncRNA prediction, we divided the snoRNAs into C/D-box, H/ACA-box and scaRNAs, miRNAs were separated into mature miRNAs (15 to 30 nts) and precursor miRNAs (39 to 6306 nts) and rRNAs were divided into short rRNAs (5S, 5.8S) and long rRNAs (23S, 25S, 28S, SSU). In general, we ended up with 10 classes for the more fine-grained prediction, each containing 5000 sequences after balancing. For short rRNA (2855 sequences) and scaRNA (1913 sequences), not enough sequences were available, which was accounted for by weighting the loss for each label individually. The datasets for each ncRNA class and subtype were split into a train and a validation dataset (90/10 split for 4500 sequences) as well as a test dataset (500 sequences) in the beginning.Aside from the sequence encoding, we also generated a graph encoding of the secondary structure using GraphProt [57], meaning secondary structure of the sequence is encoded into an input vector for classification models. The first step is to predict the secondary structure of the ncRNA sequences using the method “fasta2shrep” from [61]. The parameters used were M = 3, wins = 150 and shift = 25, meaning the secondary structure prediction was completed by creating the 3 lowest-scoring representatives of the secondary structure (shreps) of a window of 150 nts and then shifting by 25% (37 nts) and creating the next structure. All other parameters were set to default values. Based on the produced secondary structure EdeN [72] from GraphProt encodes the sequence and secondary structure information in a weighted graph-encoding vector of 32,768 features. Additionally, we performed structural encoding using the “predict_structure” method of Pysster [58], which first predicts the secondary structure using RNAfold by viennaRNA [73] and then derives the substructures and annotates each nucleotide according to the six possible substructures (F = 5′-End; T = 3′-End; S = Stem; M = Multi Loop; H = Hairpin Loop; I = Internal Loop). We then used this sequence annotation and created an arbitrary annotation that combines the nucleotide with the structure (e.g., Nucleotide A, Structure F is annotated as the character “Q”). The encoding for each combination of structure and nucleotide can be found in Table A1. For positions where the exact nucleotide is unknown but annotated according to the IUPAC code, the letter “N” is used regardless of predicted structure.

### 4.2. ML Models for ncRNA Classification

We implemented four different neural network architectures with Tensorflow 2 [74] and Keras [75] to analyze the classification of six ncRNA classes based on sequence and secondary structure information. The models are available with documentation at GitHub (https://github.com/Stegobully/merged_ncRNAclassifier, publically released on 28 March 2023). To process the whole primary sequence as input for our CNN model, we first padded all sequences to a length of 12,000 nts by appending the “_” character until the desired length was reached. Next, we encoded the sequences using ordinal encoding. Beforehand, we tested the influence of left- and right-side padding, as well as a shifted padding, but could not observe strong differences in the accuracy of the prediction. As RNA sequences may include all 16 letters of the IUPAC ambiguity code, we end up with 17 different integers, including the padding character. The CNN performs 1-dimensional convolutions and takes as input the whole ncRNA sequence after padding and encoding. To avoid different weighting of nucleotides, the encoded sequences were read into a word embedding layer consisting of 17 four-dimensional vectors. This allowed the model to learn the optimal encoding of the nucleotides, as it was already shown to be important to keep the contextual information from the sequence, such as motifs or structural information [76]. The embedding was then followed by four convolutional blocks. Each block consisted of two convolutional layers with 64 kernels each, followed by a max pooling. The kernel sizes increased with each block from 3 to 17 (block1: 3; block2: 7; block3: 11; block4: 17). The first two blocks had a max pooling size of four; the remaining two blocks had a pooling size of two. All convolutional layers used zero padding and a ReLU activation function. After the last convolutional block, a dropout layer of 50% was added, followed by a flattening layer and then a dense ReLU layer of size 10. Lastly, the output layer for the six ncRNA labels used the Softmax activation function. Training was performed using categorical cross entropy as the loss function and the optimizer was Adam [77] with a learning rate of 0.001. For the training, we used a batch size of 100 and stopped after five consecutive epochs with no improvement in validation loss. Whenever a model improved the previous lowest validation loss, it was saved, meaning the final model was the one with the overall lowest validation loss.

To handle the weighted graph encoding vectors (32,768-dimensional sparse vector) from GraphProt [57] as input for our fully-connected artificial neural network (ANN) model *GrEnc*, we performed testing on different ANN architectures, varying the number of layers and numbers of nodes in each layer. The weighted graph encoding vectors were encoded and used for training an ANN model consisting of one dense layer with 10 nodes using the ReLU activation function, followed by the dense output layer with six nodes (rRNA, tRNA, lncRNA, snRNA, snoRNA, miRNA) with the Softmax activation function. For training, the Adam optimizer with a learning rate of 0.001 and categorical cross-entropy for the loss function was set up. Training was completed with a batch size of 100 and the model trained until five consecutive epochs showed no improvement in validation loss. The model with the lowest validation loss was saved and used for testing. The combined *MncR* model had the same structure as the *SeqEnc* CNN model and the *GrEnc* ANN model in parallel with a concatenation layer that combines the dense layers with ten nodes of each model to one layer with 20 nodes. This layer was then directly followed by the output layer with six nodes (rRNA, tRNA, snRNA, snoRNA, miRNA and lncRNA) or 10 nodes (short rRNA, long rRNA, tRNA, snRNA, H/ACA-box snoRNA, C/D-box snoRNA, scaRNA, mature miRNA, precursor miRNA and lncRNA) using the Softmax activation. Training for these models was conducted without pre-assigned weights from the previous models using the Adam optimizer with a learning rate of 0.001, a batch size of 100 and categorical cross entropy as the loss function.

The optimal model hyperparameters and model architecture were gained from using grid search for filter size, dropout and learning rate for the ANN, CNN and the merged *MncR* model. For the architecture of the ANN using the structural information from GraphProt, we used combinations of one to five hidden layers with 10 to 128 neurons per layer in all combinations. For the CNN, we tested the performance using one to five convolutional blocks. After identifying the optimal number of blocks, we tested for constant, increasing and decreasing kernel sizes between 3 and 17.

Lastly, the structural encoding was first padded with the “_” character and then encoded using integers. The CNN model then had the same structure as the sequence model, but with 26 possible embeddings to account for all combinations of nucleotides as well as unannotated structures and the padding character. The architecture remained the same as the sequence model with 4 blocks of 2 convolutional layers each, followed by a max pooling layer. Best results were found with 64 kernels of size 7 for each 1D-convolutional layer. For the first two blocks, max pooling is done over 4 neighboring values and over 2 values for the last two blocks. Activation function for every convolutional layer was the ReLU function and the sequences were padded with 0 to remain the same length. After the convolutional blocks, a dropout layer of 50% was added, followed by a flattening layer, a fully connected ReLU layer with 10 nodes and lastly, the fully connected Softmax output layer. Learning rate was set to 0.001 and loss was calculated according to categorical cross-entropy. When testing ncRDeep and ncRDense, a threshold of 0.5 was chosen, as this is the default value provided in the web application.

### 4.3. Statistical Analysis

Ten-fold cross validation was performed by splitting the training set into ten balanced folds. Each fold was then used once for validation and nine times for training. Cross validation was conducted for the four model architectures to confirm the results as well as the fine-grained classes model. Model performance was evaluated according to four statistical measures, precision, recall, F1-score and Matthews correlation coefficient (MCC, [78]).

Results were assessed by analyzing overlaps within classifications using Venn diagrams and comparing above prediction scores for each class individually, where all samples of this class were considered positive and samples of all other classes as negative.

### 4.4. Benchmarking against Other ML ncRNA Classifiers

For the benchmarking of the *MncR* model against the currently best ncRNA classifier ncRDense, we used two different test datasets. The first dataset was based on our extracted 60,005 RNAcentral sequences, including miRNA, snRNA, snoRNA, lncRNA, rRNA and tRNA. From this dataset, we had 1000 sequences per class for the benchmark test set. The other dataset was used in the training and benchmark of ncRDense. The set is based on the Rfam [45] and created originally by the authors of nRC in 2017 [34]. This Rfam dataset consists of 8920 sequences from 13 different classes (5S rRNA; 5.8S rRNA; tRNA; Ribozymes; CD-Box; miRNA; Intron gp I; Intron gp II; HACA-Box; Riboswitch; IRES; Leader; scaRNA) and is split into training (6320 sequences) and testing (2600 sequences). Of these 2600 sequences, 347 were exact duplicates from the training dataset of ncRDense and an additional 9 had over 90% similarity. All 356 were removed from the test dataset.

As the comparison of the two different tools is only possible for the overlapping ncRNA classes, we first removed from the RNAcentral test set all sequences with >90% similarity to any training sequence from ncRDense and all sequences from the Rfam test dataset with >90% similarity to the training sequences from the *MncR* model. Additionally, only six classes occurred in both models, meaning exclusion of the sequences with labels for Ribozymes, Intron gpI/gpII, Riboswitch, IRES and Leader (Rfam), lncRNAs and snRNAs (RNAcentral). The classes H/ACA-box, C/D-box and scaRNA from Rfam were labeled as snoRNA for the testing of *MncR* and vice versa. Furthermore, the labels of 5.8S and 5S rRNA were changed to rRNA and for the RNAcentral dataset, only sequences of 5S and 5.8S rRNA were included from the rRNA sequences. As the training dataset of ncRDense did not contain any mature miRNAs, we also excluded them in the test dataset of RNAcentral. Lastly, the input mask of ncRDense only allows sequences of a maximum length of 750 nts, which resulted in a remaining benchmark dataset of Rfam containing 1125 sequences (184 miRNA; 347 rRNA; 401 snoRNA; 193 tRNA) and RNAcentral dataset containing 2737 sequences (523 miRNA; 269 rRNA; 952 snoRNA; 993 tRNAs). To test the Rfam set on our model, we first pre-processed the sequences the same way as for the RNAcentral set (padding to 12,000 characters, encoding the sequence ordinally, deriving graph features using GraphProt) as well as relabeling the sequences to the ncRNA types. The unprocessed sequences were uploaded into ncRDense in FASTA format, split into 2 files, as ncRDense only allows for a maximum of 1000 sequences per input. The results were saved and fine-grained labels were relabeled to the ncRNA class, meaning a CD-Box sequence classified as HACA-Box by ncRDense was deemed as correctly classified. Sequences predicted as a label not present in the other model were deemed incorrect. The results were then analyzed for overall precision, recall, F1-score and MCC, as well as class-wise F1-scores. For the RNAcentral set, the eligible sequences were uploaded in 3 different FASTA files to the web tool and then analyzed the same way as the Rfam sequences. For the *MncR* model, we simply filtered the already tested sequences from the comparison between *GrEnc*, *SeqEnc*, *StrEnc* and *MncR* to the ones eligible for ncRDense.

## Figures and Tables

**Figure 1 ijms-24-08884-f001:**
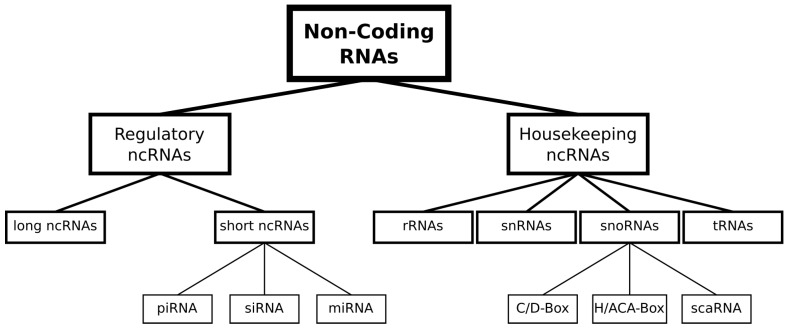
Non-coding RNA classes and grouping. Hierarchical tree structure of different ncRNA classes grouped into regulatory and housekeeping ncRNAs. Grouping of ncRNA classes is based on Xiang-Dong Fu [4].

**Figure 2 ijms-24-08884-f002:**
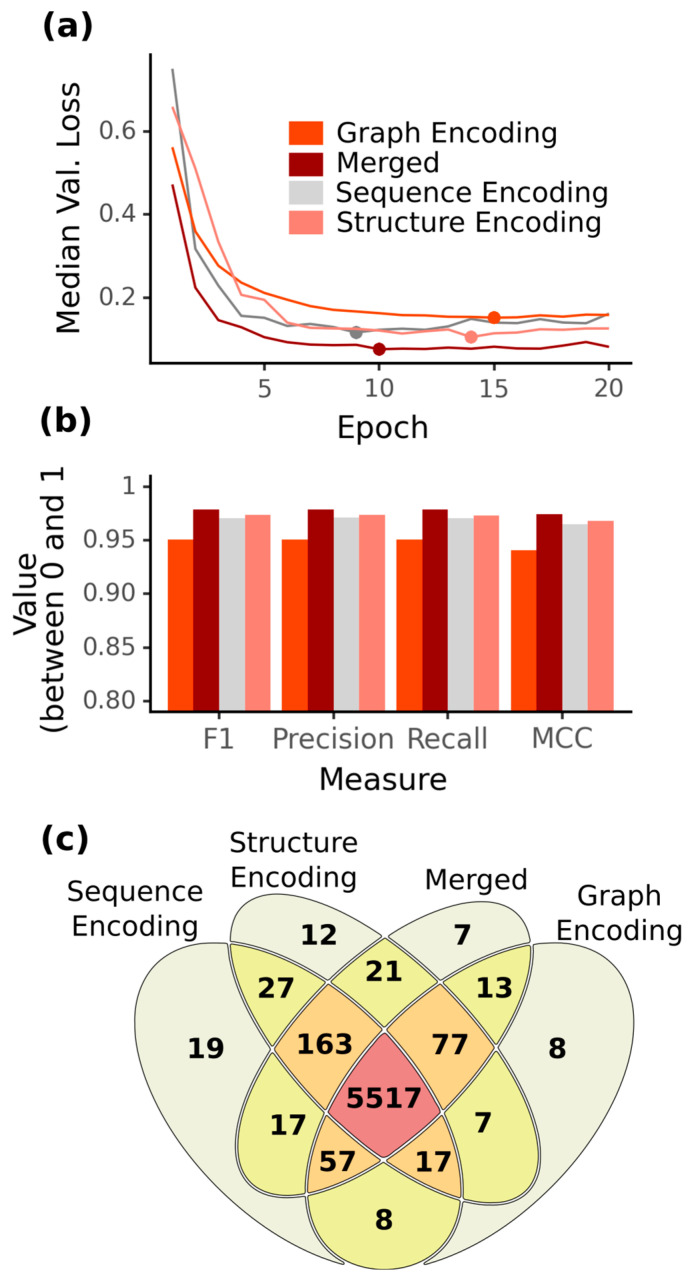
Quality scores for the investigated ncRNA classification models. (**a**): For each model, (graph encoding (*GrEnc*): red; merged (*MncR)*: dark red; sequence encoding (*SeqEnc*): light grey; structure encoding (*StrEnc*): light red) the median validation loss for each epoch during cross-validation is plotted (line). The dot marks the lowest median validation loss for each model. (**b**): For the different models, the measures of F1-score, precision, recall and MCC (Matthews correlation coefficient) on the RNAcentral test set are visualized as values between 0 and 1. MCC can theoretically take up values below 0, but only in the case of mislabeled training or test samples. (**c**): Venn diagram showing the overlap between correct classifications for each model. The total number of samples was 6001. Yellow indicates an overlap of two models correctly predicting the sequence; orange indicates an overlap of three models; red indicates that all models predicted the sample correctly.

**Figure 3 ijms-24-08884-f003:**
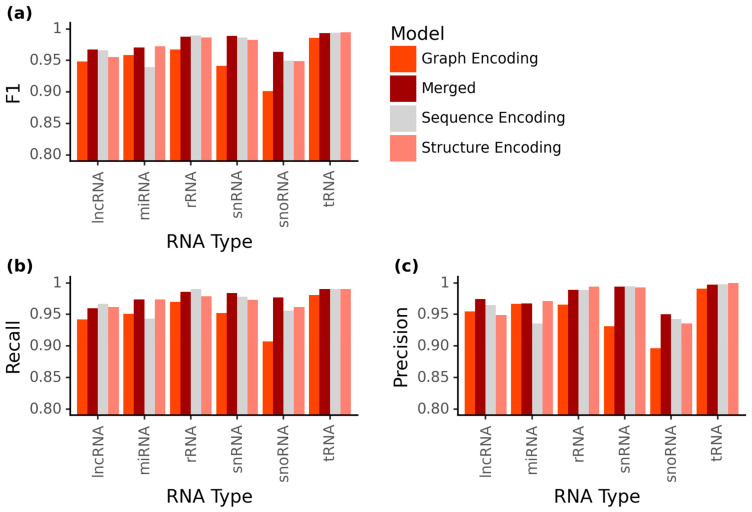
ncRNA class-wise prediction scores. Our four different models (graph encoding (*GrEnc*): red; merged (*MncR)*: dark red; sequence encoding (*SeqEnc*): light grey; structure encoding (*StrEnc*): light red) and their prediction scores for each class on the RNAcentral test set compared based on (**a**) F1-score, (**b**) recall and (**c**) precision.

**Figure 4 ijms-24-08884-f004:**
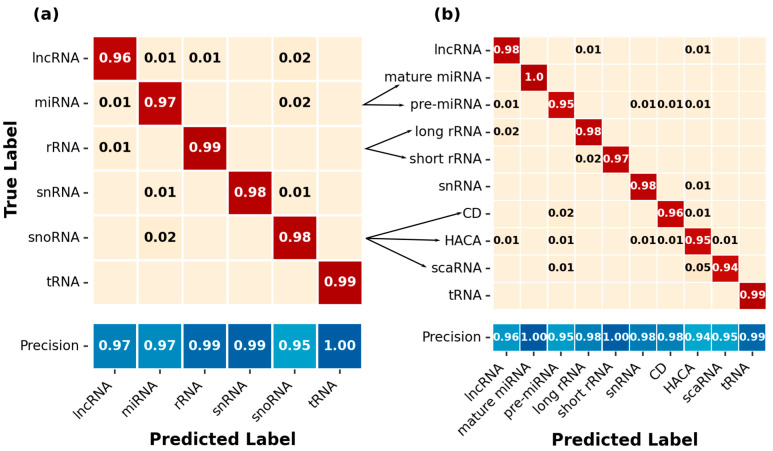
Classification and precision changes by subtype prediction in the *MncR* ML model for general classes and fine-grained classes. Confusion matrices (normalized to each row) for (**a**) general classes *MncR* and (**b**) fine-grained classes *MncR* are shown with true labels as rows and predicted labels as columns. The fraction of predictions is indicated by the number in each square as well as the color gradient (low value: beige; high value: dark red). Values below 0.005 are omitted from the matrices. Values on the main diagonal in each matrix are equivalent to the recall for this class. Precision values (between 0 and 1) for each ncRNA class are visualized by the blue color gradient below the confusion matrices.

**Figure 5 ijms-24-08884-f005:**
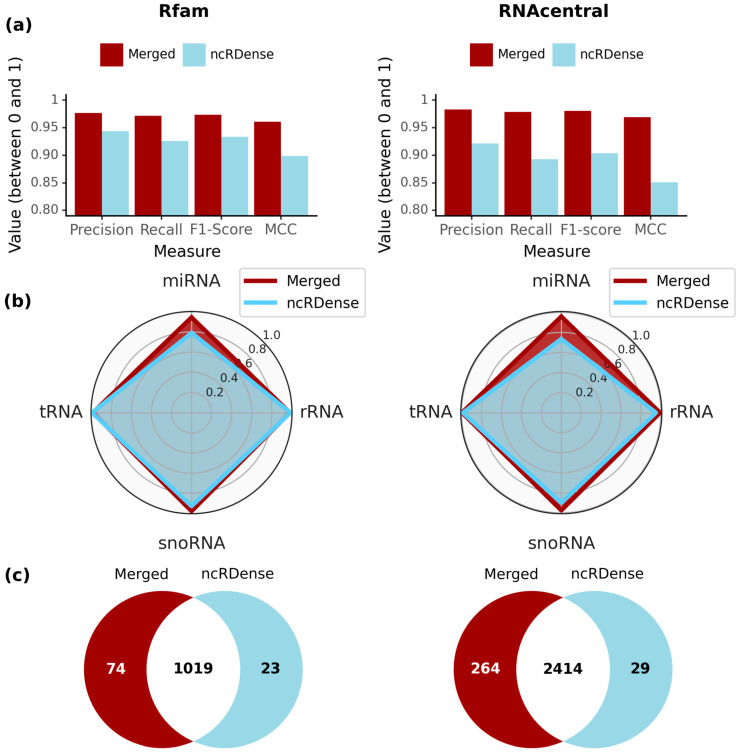
Benchmark between Merged (*MncR)* and ncRDense on the Rfam and RNAcentral benchmark sets. The models (*MncR*: dark shades; ncRDense: light shades) are compared on the eligible sequences from the Rfam (left, blue) test set and the RNAcentral (right, green) test set (**a**) by evaluating overall precision, recall, F1-score and MCC (all values between 0 and 1) and (**b**) by comparing class-wise F1-scores for each ncRNA class. (**c**) Venn diagram of the correctly predicted sequences from the Rfam test set (1125 sequences) and RNAcentral test set (2737 sequences).

**Table 1 ijms-24-08884-t001:** Overview of false classifications of the different ML models. All numbers are absolute numbers from the RNAcentral test set. The total number of sequences in the test set is 6001. The rows show the six ncRNA classes and the sum. The columns state the overall wrongly assigned sequences in at least one model or all models and the detailed information about wrongly predicted sequences from the *MncR* model.

ncRNA Class	Min. One Model Wrong	*MncR* Wrong	All Models Wrong	Only *MncR* Wrong
lncRNA	87	40	12	6
miRNA	107	26	4	3
rRNA	50	14	1	2
snRNA	76	16	2	2
snoRNA	136	23	6	3
tRNA	28	10	4	1
Sum	484	129	29	17

**Table 2 ijms-24-08884-t002:** Number of sequences per ncRNA class and subclass. Total number of sequences: 60,005. For values marked with *, this is the maximum number of clustered sequences available at time of download. Subtype may be unspecified either due to a large number of unknown or unannotated samples, or because too many similar subtypes exist.

ncRNA Class	Subtype	Count
lncRNA	unspecified	10,000
miRNA	Mature miRNA	5000
	Precursor miRNA	5000
rRNA	23S	1435
	25S	1409 *
	28S	1435
	5.8S	1420 *
	5S	1435
	Small Subunit (SSU)	1435
	Mitochondrial (mt)	1435
snRNA	unspecified	10,000
snoRNA	C/D-Box	4044
	H/ACA-Box	4044
	scaRNA	1913 *
tRNA	unspecified	10,000

## Data Availability

The training and test FASTA files as well as machine learning models used for the analysis can be downloaded on the github repository (https://github.com/Stegobully/merged_ncRNAclassifier) (published on 28 March 2023). Graph and structure encoding files corresponding to the FASTA files are available on request.

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
