# Peer review of "MncR: Late Integration Machine Learning Model for Classification of ncRNA Classes Using Sequence and Structural Encoding"

_ijms, 2023, doi:10.3390/ijms24108884_

Round 1
Reviewer 1 Report
This is a study that uses deep learning to classify different types of ncRNA. The study is methodologically sound, and the manuscript is clearly written and organized. It should be of interest to the ncRNA community. Some caveats:
1. The need (in the broader biomedical context) for accurate ncRNA classification is not sufficiently shown. Yes, ncRNAs are important. But what are the actual benefits (examples?) of classification?
2. The performance improvements appear to be incremental, perhaps well within the range of the prior (simpler, and presumably more computationally efficient) methodologies' parameter and hyperparameter tuning.
3 (major): it is unclear to this reviewer why there was a need to join *all* classes of ncRNA in the same single analysis framework. From the research community's viewpoint, the motivation and utility are not obvious. Moreover, stratifying into, say, short and long types might significantly increase both performance and computational efficiency. On a related note, "_" padding can be problematic --- have the authors investigated and cross-tested the alternatives?
4 (minor): CNN is *also* an ANN. Perhaps the authors meant the "default" ANN, which is a basic feed-forward multilayer perception with backprop?
Author Response
Dear Ms. Kankanok Wongnavee, Reviewer of the manuscript and Academic Editor of IJMS,
We would like to resubmit the minor revision of the manuscript “MncR: Late integration Machine Learning model for classification of ncRNA classes using sequence and structural encoding” (ijms-2343934). We rephrased parts of the manuscript and added content concerning the comments from the two reviewers as well as the academic editor. Especially, the introduction and discussion part including the tools ncDC, NCodR, mSRFR and ncRNLP and their comparison to MncR have been added. Furthermore, we added an additional supplementary figure S3 in the supplements zip-file and updated there the figure and table legends as well as changed the coloring of Figure 5. The figure is also updated by track changes in the manuscript and the zip-file with all supplementary files is also now in the uploaded zip-file Revision_ijms_2343934. The full point-by-point response to the comments is added in the response letter copied in the free text field as well as in the zip-file. As recommended all changes are marked by track changes in the main word manuscript.
Comments from Reviewer 1:
- This is a study that uses deep learning to classify different types of ncRNA. The study is methodologically sound, and the manuscript is clearly written and organized. It should be of interest to the ncRNA community. Some caveats: The need (in the broader biomedical context) for accurate ncRNA classification is not sufficiently shown. Yes, ncRNAs are important. But what are the actual benefits (examples?) of classification?
Response: Thank you for the valuable comment. We agree with the reviewer and added in the introduction the importance of ncRNA classification now by also adding four references to show that in respect of the new sequencing techniques the classification of such reads in sense of ncRNAs can help to save time and money in the sense of first regulatory functionality assignments by differing between potential siRNAs, miRNAs or tiRNAs. The manuscript includes now: “Since 2003 more and more publications focus on the identification of new classes of ncRNAs based on either sequential, structural or functional information, for which reason “computational RNomics” for genome wide annotation of RNAs became important [25]. Classification of short and long ncRNA classes can help to further characterize potential new ncRNAs found in high-throughput sequencing [26] and already assign them in a putative functional context without time consuming in vitro or in vivo experiments. Besides the long and small ncRNA categories the assignment of new ncRNAs to classes like tiRNAs, siRNAs or miRNAs allows a prefiltering for the regulatory functionality in relation to diseases like cancer [16] as well as their putative functionality for individualized therapies [27].”
- The performance improvements appear to be incremental, perhaps well within the range of the prior (simpler, and presumably more computationally efficient) methodologies' parameter and hyperparameter tuning.
Response: In regards to this excellent comment we now added in the methods parts more clearly that we optimized the ML-models independently for all hyperparameter and also tested several different architectures beforehand to find the best model: “The optimal model hyperparameters and model architecture were gained from using grid search for filter size, dropout and learning rate for the ANN, CNN and the merged MncR model. For the architecture of the ANN using the structural information from GraphProt we used combinations of one to five hidden layers with 10 to 128 neurons per layer in all combinations. For the CNN we tested the performance using one to five convolutional blocks. After identifying the optimal amount of blocks we tested for constant, increasing and decreasing kernel sizes between 3 and 17.” For this reason we would conclude, that we cannot increase the accuracy of the sequence or structure only models by parameter tuning as this has been done already. This means, the merged model definitely has extracted more information from the combination of both level structure and sequence in our opinion.
- (major): it is unclear to this reviewer why there was a need to join *all* classes of ncRNA in the same single analysis framework. From the research community's viewpoint, the motivation and utility are not obvious. Moreover, stratifying into, say, short and long types might significantly increase both performance and computational efficiency. On a related note, "_" padding can be problematic --- have the authors investigated and cross-tested the alternatives?
Response: We thank the reviewer for this comment and added now in the methods part a clearer definition of our embedding layer to decrease the possibility of learning artifacts from the padding. Furthermore, we tested both left and right padding as well as a shifted padding approach but did not see any differences in the accuracy: “Next, we encoded the sequences using ordinal encoding. Beforehand we tested the influence of left- and right-side padding as well as a shifted padding but could not observe strong differences in the accuracy of the prediction.” As RNA sequences may include all 16 letters of the IUPAC ambiguity code, we end up with 17 different integers including the padding character. The CNN performs 1-dimensional convolutions and takes as input the whole ncRNA sequence after padding and encoding. To avoid different weighting of nucleotides, the encoded sequences are read in into an word embedding layer consisting of 17 four-dimensional vectors. This allows the model to learn the optimal encoding of the nucleotides as it was already shown to be important to keep the contextual information from the sequence like motifs or structural information [77].” In regards of the second comment within here the separation of long and small ncRNAs in separate models would not really allow a higher accuracy but mainly a higher false positive rate as many of the classes are not available for the prediction. Also, we would argue based on our results that the accuracy in our multiclass-model is already very high in the combination of small and long ncRNAs. In addition, the separation into several models would increase the chance of false positives by the intrinsic problem that all sequences will be classified in a model and by the lack of specific groups this means they will be falsely classified instead of not classified. As there are many tools available for the discrimination of two ncRNA classes or ncRNAs vs. mRNAs we can be sure that the addition of small ncRNAs will be wrongly assigned as they have to get a prediction. Based on this the combination of long and small ncRNAs can create a bias in classifying all long sequences as lncRNAs and short sequences as miRNAs what we could not observe in our model. Also, we tested this with the inclusion of mature and pre-miRNAs as well as splitting long and short rRNAs and could not really observe a better accuracy. We also elaborated this more in the manuscript in the introduction: “At stage ncRNA predictors have a maximum input length of 750 nts, which is not allowing to predict ncRNA classes like rRNAs and miRNAs (pri-miRNAs >10,000nts [47]) including long and small ncRNA sequences leading to false positive classifications, even when using separated machine learning models specialized on long or small ncRNAs. Furthermore, the class of lncRNAs [48] can exceed the length of 750 nts but in the current definition the minimum length is 200 nts, which is also problematic in covering this class in a specific long or small ncRNA classifier.” Further we added a paragraph discussing this point together with the reviewer comment 1 from reviewer to in the discussion (See comment response 1 of Reviewer2).
- (minor): CNN is *also* an ANN. Perhaps the authors meant the "default" ANN, which is a basic feed-forward multilayer perception with backprop?
Response: We thank the reviewer for the careful correction of our manuscript. To clarify this uncertainty from our side we now specified our usage of ANN the following: “By late integration our ML-model MncR (Merged_ncRNAclassifier) combining a CNN for sequence encoding information and a fully connected feed-forward artificial neural network (from now on in this article abbreviated with ANN) based on weighted graph encoding was benchmarked based on overall and single class predictions.” and “To get insights into the overall prediction capacity of sequence and structure information as well as their combination we implemented and optimized four ML-models using fully connected feed-forward ANNs and CNNs as well as late integration of both to predict six ncRNA classes miRNA, snRNA, snoRNA, tRNA, rRNA, lncRNA.”
Comments from Reviewer 2:
The authors developed a recognition method for non-coding RNA based on the primary and secondary structural characteristics of RNA. This is a very meaningful work. Here are a few issues to focus on:
- This is a multi-classification problem. From the results of the author's research, the effect is indeed good; But from Figure 4 and Table 2, the author solves this problem well? Could you elaborate and discuss it.
Response: We agree with the author that we added results in Figure 4 and Table 2 according to the adding of more sub-classes of snoRNAs for example but did not discuss them or commented on them in the results. For this reason we added the following in the discussion section of the manuscript: “Additionally, we split the ncRNA classes rRNA and miRNA into long and short to see if this approach has an influence on the prediction accuracy. This approach is in principle comparable to using two distinct classifiers for long and small ncRNAs with the advantage to give the ML-model the chance to also select from the additional ncRNA classes. For the rRNA we could observe no improvement in the split of long and small rRNAs but a slight decrease in the overall performance. For the miRNAs and pre-miRNAs split we observed a decreased performance in the detection of pre-miRNAs and increased performance on miRNAs. Overall, the sequence length based categorization for miRNAs as pre-(cursor) and mature miRNAs as well as for rRNAs as long and short rRNAs showed no real improvement, if not even a slight decrease in prediction capacity. Furthermore, the inclusion of small and long ncRNA classes even including the embedding showed no decrease in the overall prediction accuracy with F1-scores above 0.95 (Figure 2 and Figure 4). Focusing on the falsely assigned sequences per ncRNA class, we observed that the few false positives in the MncR model were irrespective of the length of the ncRNA sequence.”
- The colour style of Figure 5 is inconsistent with Figure 2 and 3.
Response: Thank you for this comment. We agree with the author and now changed the colour in Figure 5 according to the colour scheme from the figures before. For this reason the merged model is now still red and the ncRDense is now blue.
- Can you discuss and explain whether the primary structure is more important or the secondary structure is more important for RNA identification?
Response: Thank you for this valid point. We now added in the discussion also in relation to the mentioned comment before a small paragraph discussing the importance of primary and secondary structure and the addition of more classes. The manuscript discussion now has the following paragraph: “Focusing only on the overall prediction accuracy can lead to the conclusion that the sequence information alone performs better than the structural information (Figure 2). Beside the overall prediction accuracy, we observed different strengths and weaknesses of the single models regarding the single ncRNA classes (Figure 3). In general, we observed for five of the six ncRNA classes a better performance on the sequence alone, only the miRNAs were more accurately predicted using the structural information.” and “Based on our results the hypothesis that ncRNA class prediction with a specific structure like snoRNAs benefit from using just structural information could not be confirmed. Moreover, the single information (sequence or structure) show tendencies to improve specific ncRNA classes and the combination in the merged MncR combines both strengths.”
Comments from the Editor:
- " Simm's work concerns an important and innovative field of research. The new model, MncR, is claimed to be superior to ncRDense and the authors boast that it can predict long ncRNA classes up to 12,000 nts.
The authors should discuss and compare their model with those recently published by Ruengjitchatchawalya (https://biodatamining.biomedcentral.com/articles/10.1186/s13040-022-00291-0), Jha (https://link.springer.com/article/10.1007/s41870-022-01064-y), Bahadur (https://www.cambridge.org/core/journals/quantitative-plant-biology/article/ncodr-a-multiclass-support-vector-machine- classification-to-distinguish-noncoding-rnas-in-viridiplantae/39C9F5328BBF3F450BDD6C3A52D817EC), and, especially, that of Liang (https://www.nature.com/articles/s41598-022-22082-7#Sec22) for long non‑coding RNA detection."
Response: Thank you for this valid point. We now added in the discussion the mentioned literature and compared their results to our approach in respect to explaining the differences and unique features of MncR in comparison to the tools LncDC, NCodR, mSRFR and ncRNLP. In the introduction we added the three missing tools: “As research into ncRNAs in medical applications is currently mainly focused on specific ncRNA classes as well as individual ncRNAs, it is important to develop robust approaches for ncRNA classification to identify putative new candidates and allow a clear assignment. Since 2003 more and more publications focus on the identification of new classes of ncRNAs based on either sequential, structural or functional information, for which reason “computational RNomics” for genome wide annotation of RNAs became important [25]. Classification of short and long ncRNA classes can help to further characterize potential new ncRNAs found in high-throughput sequencing [26] and already assign them in a putative functional context without time consuming in vitro or in vivo experiments. Besides the long and small ncRNA categories the assignment of new ncRNAs to classes like tiRNAs, siRNAs or miRNAs allows a prefiltering for the regulatory functionality in relation to diseases like cancer [16] as well as their putative functionality for individualized therapies [27]. The development of next generation sequencing (NGS) [28] applications has led to a very fast increase in the possibilities to generate large amounts of data pertaining to expressed RNAs and partial RNA sequences. Therefore, machine learning (ML) approaches have been developed to improve the classification of RNA sequences based on primary sequence and secondary structure. There are a variety of approaches available for the differentiation between coding and non-coding RNA including CPC2 [29] as well as tools for the identification of individual ncRNA classes including LncDC [30] for differentiating lncRNAs from mRNAs. More recently, the rapid amount of increasing ncRNA sequences due to RNA-Seq experiments allowed the development of tools focusing their training on just a specific set of species like microalagae (mSRFR [31]) or a taxonomic rank like Viridiplantae (NCodR [32]) to reach higher accuracy. Especially, the differentiation between multiple ncRNA classes within one sample may be of interest among a set of sequences, as this has the potential to improve detection of ncRNAs in non-model organism genomes as well as annotation of contigs of RNA-Seq samples in the future. For this reason, the classification of multi-class ncRNAs is of major importance and has evolved from multiple sequence alignment, over simple machine learning classifiers (GraPPLE [33], RNAcon [34]) and deep learning artificial intelligence (AI) approaches including nRC [35], ncRFP [36], ncRDeep [37] and ncRDense [38] up to natural language processing methods (NLP) like ncRNLP [39].” Further, we checked the performance of plant sequence classification within our tool in comparison to NCodR (See newly created Supplementary Figure 3). The following paragraph in the discussion was added: “In contrast to tools like LncDC [30] the focus of MncR is not to discriminate between protein coding mRNAs and specific ncRNA classes like lncRNAs. Such tools are important to pre-filter datasets of RNA-seq sequences to exclude mRNAs as those have partly similar features to lncRNAs or to identify putative novel lncRNAs. Often the training to discriminate between mRNA and ncRNA has to be done individually for specific species like human in LncDC [30] or taxonomic classes like microalgae in mSRFR [31]. The approach of mSRFR showed clearly that specific distinctions of sequence types between different kingdoms exist and have to be present in the training dataset to be learned by the ML-model. The other universal sequence classifiers like MncR or ncRDense [38] want to achieve the detection of ncRNAs irrespective of special features of distinct species or taxonomic kingdoms. Tools like mSRFR [31] or NCodR [32] focusing on a specific set of species increases the sensitivity for distinct features like additional variants of snRNAs in plants, which can be very short in length [71], but also decrease the possibility to be generalized and increase the chance of overfitting. In the case of NCodR the publication showed a drastic increase in the F1-score by 30% compared to ncRDeep solely trained on Viridiplantae sequences [32]. To test MncR on a plant specific dataset we created a test set with 996 sequences (labelled with the current RNAcentral annotation) from the ~500.000 plant sequences of NCodR and classified them with MncR achieving an F1-score of 0.85 (Suppl. Figure S3). As MncR is trained on a dataset including ~11% of plant sequences the model was still able to achieve high recall values for all types of plant ncRNAs (> 0.87) with the exception of snRNA (~0.40). Besides the internal comparison of different models using different input sources, we compared our MncR model to the currently best multi-class ncRNA classifier ncRDense [38], which is also based on deep learning, As ncRDense outperforms the other tools ncRFP and nRC by at least 0.17 (F1-score) and ncRDeep by 0.07 we wanted to compare our MncR model with ncRDense. Therefore, we used the classic Rfam test set created from the developers of nRC [35] and adjusted it based on the common ncRNA classes. MncR (F1-score: 0.9738) achieved higher accuracies than ncRDense with increased precision (+0.033), recall (+0.045) and F1-score (+0.04) (Figure 5). It was able to correctly classify 74 sequences, which ncRDense failed to classify, while ncRDense only correctly classified 23 sequences that our model could not correctly predict. The scores for ncRDense differ slightly from their publication, as we had to adapt the test set to not include any duplicates to either of the training sets of the tested models. In 2023 a new tool based on NLP called ncRNLP [39] trained and tested on the same Rfam dataset like ncRDeep and ncRDense was released but benchmarked only against ncRDeep. Both publications of ncRDense and ncRNLP compare themselves to ncRDeep (F1-score: ~0.88) and got similar benchmark results resulting in an improvement of the F1-score by 0.07 (ncRDense) to 0.09 (ncRNLP). Relating these benchmarks to our results allows to conclude that our ML-model MncR also performed slightly better than ncRNLP on the Rfam dataset. By checking the benchmark datasets, we observed that 347 sequences of the original Rfam benchmark dataset from 2017 are present in both the training and the test set of ncRDense, of which 131 belong to the four ncRNA classes we compared. When only testing the sequences of overlapping ncRNA classes, our MncR model overall outperforms ncRDense (Figure 5), mainly because of an increased accuracy in miRNA prediction. Even more drastic is this result using our newly designed RNAcentral benchmark dataset. The biggest difference in the results can be found for miRNA but also for the other ncRNA classes tRNA, rRNA and snoRNA we achieved better accuracies for our MncR model.”
Thank you very much for considering our revised manuscript and with kindest regards,
Stefan Simm

Reviewer 2 Report
The authors developed a recognition method for non-coding RNA based on the primary and secondary structural characteristics of RNA. This is a very meaningful work. Here are a few issues to focus on:
1. This is a multi-classification problem. From the results of the author's research, the effect is indeed good; But from Figure 4 and Table 2, the author solves this problem well? Could you elaborate and discuss it.
2. The color style of Figure 5 is inconsistent with Figure 2 and 3.
3. Can you discuss and explain whether the primary structure is more important or the secondary structure is more important for RNA identification?
Author Response

(The authors gave the same response as above.)
